# SCALE-ADAPTER: REVERSED DISTILLATION ADAPTER FOR EFFICIENT TRAINING OF LARGE VIDEO DIFFUSION MODELS

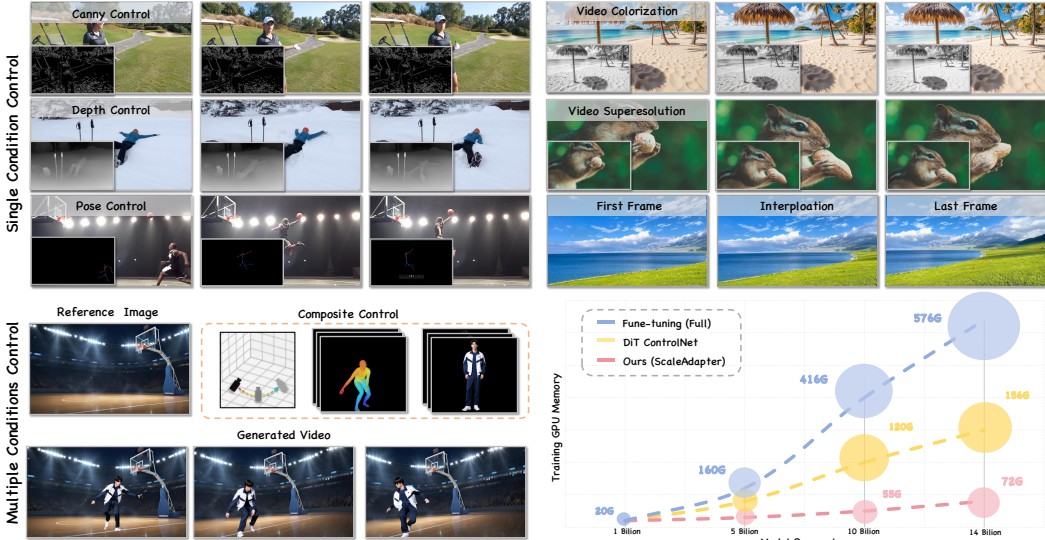

Figure 1: **Visual results of Scale-Adapter**. We propose the Scale-Adapter, a plug-and-play adapter that enables efficient training and flexible extension to diverse conditions, including both single conditions (e.g., canny, pose, depth) and multiple condition compositions (e.g., camera trajectory, image background, human motion) with minimal GPU consumption.

## ABSTRACT

We propose Scale-Adapter, a plug-and-play adapter designed to efficiently integrate conditional knowledge from smaller adapted models into large video diffusion transformers. Existing controllable video DiT methods face critical challenges: full fine-tuning of billion-parameter models is extremely expensive, while cascaded ControlNets introduce substantial parameter overhead and exhibit limited flexibility for novel multi-condition compositions. To overcome these issues, Scale-Adapter introduces a novel reversed distillation method, enabling large video diffusion models to inherit precise control capabilities from smaller, efficiently-tuned video diffusion models, completely eliminating full fine-tuning. Moreover, recognizing the intrinsic relationships between different conditions, we replace the cascaded ControlNet design with a *Mixture of Condition Experts* (MCE) layer. This structure dynamically routes diverse conditional inputs within a unified architecture, supporting both single-condition control and multiple condition combinations without additional training cost. To achieve cross-scale knowledge transfer, we further develop a *Feature Propagation Module* to ensure efficient and temporally consistent feature propagation across video frames. Experiments demonstrate that Scale-Adapter enables high-fidelity multiple condition video synthesis, making advanced controllable video generation feasible on low-resource hardware and establishing a new efficiency standard for the field.

## 1 INTRODUCTION

Recent advances in Diffusion Transformers (DiTs) have revolutionized high-fidelity video synthesis driven by text prompts, enabling unprecedented visual quality (Peebles & Xie, 2023). However, generating spatiotemporally coherent content solely through text remains challenging due to the lack of guidance for fine-grained structural details (*e.g.*, object layouts, motion trajectories). To address this, control conditions, such as bounding boxes, segmentation maps, and depth maps, have been integrated into diffusion frameworks. Notably, ControlNet (Zhang et al., 2023) and T2I-Adapter (Mou et al., 2023) have emerged as dominant solutions, extending Stable Diffusion (Rombach et al., 2021; Podell et al., 2024; He et al., 2022) with lightweight adapters to support diverse input conditions, fostering broad adoption in controllable image generation. These methods enhance the conditional control capability of models by freezing the parameters of the main image generation network and introducing additional trainable parameters.

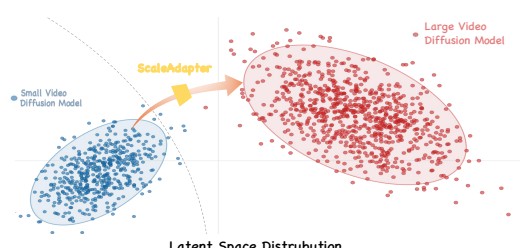

When handling multiple conditional inputs, a common approach is to introduce additional ControlNets, each specialized for a specific condition (Sun et al., 2025). However, this strategy leads to a linear increase in model parameters and requires repeated training processes for each new condition, resulting in significant computational overhead. Recent efforts Lin et al. (2024) attempt to mitigate this by incorporating adapter modules and routers to combine multiple pre-trained ControlNets for image diffusion models. Despite these improvements, such methods still rely heavily on existing pre-trained ControlNets, require substantial training resources, and exhibit limited extensibility to novel conditions.

Figure 2: **The illustration of latent feature distribution transformation.**

Despite advancements in adaptive condition generation for video synthesis, current frameworks still face three critical challenges: (1) **Training Efficiency**: Fine-tuning ControlNets for DiT-based video diffusion models necessitates an enormous number of parameters. Incorporating a new conditional control typically requires about 0.5 billion parameters and significant computational resources (exceeding 48 GPU hours) for high-quality datasets. This imposes a substantial resource burden, compounded by the fact that state-of-the-art video models now have over 14 billion parameters.

(2) **Inflexible Multi-Condition Fusion**: Prior works have extended conditional control from image generation to video generation tasks. While image-generation ControlNet architectures have been adapted for video tasks, they fail to address specific video control requirements, such as camera motion, background, or character features. Crucially, combining multiple conditions often relies on cascading specialized ControlNets. For a typical multiple conditions setup, it needs to train each separate ControlNet to learn each and equip it with a large base model (e.g., the Wan2.1 14B model), and the conditions cannot be dynamically integrated. (3) **Limited Condition Consistency**: Image-conditioned adapters often fail to maintain temporal and conditional coherence when

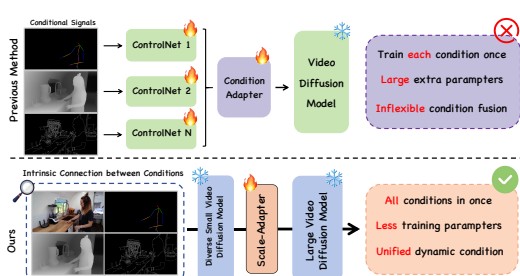

Figure 3: **Motivation of our method**. Compared to previous methods, our framework only requires training adapters in low-resource environments to support diverse conditions, eliminating redundant training efforts.

applied to video generation, resulting in visible artifacts, including frame flickering and unstable content such as fluctuating characters or backgrounds. Although some extended approaches introduce temporal convolutions and linear projection layers into ControlNets, they still do not explicitly model the spatial correspondence and time-step alignment between conditioning features. This fundamental limitation necessitates large volumes of training data and extended training time, while still failing to ensure stable and controllable generation outcomes. At the same time, as shown in Figure 2, we noticed that small and large models within the same architecture family exhibit strong feature

similarity (detail in Sec . A in the appendix ), allowing knowledge, particularly in latent space, to be efficiently transferred from a fine-tuned small model to a large pretrained foundation model. Moreover, we note that low-resource fine-tuning can equip small models with richer and more diverse conditional control abilities than those achieved by single-condition ControlNet adaptations.

To overcome these limitations, we propose a unified framework centered on three innovations: First, we introduce Scale-Adapter, a novel and efficient method developed with minimal training overhead for conditional video synthesis. Instead of training a full video model, it enables synergistic collaboration between large foundation models and small video diffusion models, as shown in Figure 3. Unlike previous methods, we have found that small models and large models with the same architecture can achieve cross-scale knowledge bridging. Second, we design a *Mixture of Condition Experts* (MCE) that concurrently processes heterogeneous input conditions for video generation tasks in a single forward pass, and we find the intrinsic relationships between different visual conditions, thereby eliminating repeated training cycles. Unlike previous adapter-based methods, MCE employs dynamic routing to activate relevant experts for conditions, leveraging inter-condition synergies learned during joint training. As shown in Figure 6, this results in fewer parameters compared to Multi-ControlNet (Sun et al., 2025). Third, we develop a Feature Propagation Module to ensure feature reversed propagation. Conditional features from the adapter are scaled and projected into each video DiT block, aligning the injected controls with the base model's priors. Our approach reduces condition-specific video training costs, supports dynamic composition of novel conditions, and cuts parameter overhead versus ControlNet and adapters, setting a new efficiency framework for controllable video generation. Our contributions are summarized as follows:

- We present Scale-Adapter, a plug-and-play adapter to transfer the controllable knowledge from small-parameter video models to large-parameter models efficiently.

- Technically, we first design a *Mixture of Condition Experts* (MCE) layer that covers various control signals with dynamic expert routing. It shows the ability to adapt to unseen conditions by learning the intrinsic relationships between different visual conditions.

- To achieve reversed condition distillation, we develop a *Feature Propagation Module* that efficiently ensures condition coherence during feature transfer in the denoising stage.

## 2 RELATED WORK

**Video Diffusion Model** Generative modeling has propelled remarkable advancements in large-scale video models, with diffusion-based frameworks emerging as a prominent area of development (OpenAI, 2024; Gafni et al., 2022; Chen et al., 2023; 2024a; Dai et al., 2023b; Guo et al., 2024; Long et al., 2024; Goodfellow et al., 2020; Khachatryan et al., 2023). A large number of diffusion-based video generation approaches are built upon the Stable Diffusion (Rombach et al., 2021; Blattmann et al., 2023; He et al., 2022; Xing et al., 2023), encompassing three fundamental components: an autoencoder that transforms raw videos into a compact latent space (van den Oord et al., 2018); a text encoder tasked with extracting text embeddings (Raffel et al., 2023); and a neural network, optimized through diffusion processes, (Ho et al., 2020;?; Qiu et al., 2023) that learns the distribution characteristics of these video latents. In terms of architectural design, the U-Net, originally devised for image generation tasks, has been adapted to video generation by integrating temporal dimensions. Notably, Diffusion Transformers (DiTs)(Peebles & Xie, 2023; Guo et al., 2023; Chen et al., 2023; Wang et al., 2024b; Peebles & Xie, 2023), which employ exclusively transformer blocks, have exhibited superior performance over U-Net architectures in the domain of visual generation.

**Controllable Generation in Diffusion Models** The remarkable success of diffusion models (Wan et al., 2025; Li et al., 2023b; Hong et al., 2022; Li et al., 2017; Ho et al., 2022; Zhou et al., 2022; Wang et al., 2024b; Singer et al., 2023; Chen et al., 2023) has spurred substantial interest in controllable video generation. To address the need for fine-grained control over diffusion-based synthesis, researchers have explored a wide range of conditional inputs, including depth maps, Canny edges, reference images, and multimodal combinations. However, the computational cost of full-parameter fine-tuning for each new condition has driven the development of parameter-efficient adaptation methods. Notable approaches in this domain include ControlNet (Zhang et al., 2023) and T2I-Adapter (Mou et al., 2023), which enable pretrained diffusion models to incorporate additional conditional signals through lightweight trainable branches. These methods effectively balance

expressiveness and efficiency. UniControl (Qin et al., 2023) introduces a MoE-style Adapter and a Task-aware HyperNet to support diverse tasks within a single model. However, its task adaptation mechanism is designed for text instructions and does not explicitly model relationships between tasks and conditions. Multi-ControlNet (Sun et al., 2025), which enables composite control, suffers from isolated branches that limit composability. Uni-ControlNet (Zhao et al., 2023) addresses this by grouping conditions into local and global controls, supporting composable control within a single model. Nevertheless, its inability to maintain consistency across frames hinders its applicability to video generation. Inspired by ControlNet, DiT-ControlNet (Denis, 2025) incorporates zero modules into the DiT architecture to learn new conditions without training the backbone model. While effective, this approach incurs significant training overhead. Ctrl-Adapter (Lin et al., 2024) injects latent feature maps into video generation models using image ControlNets and adapters inserted into each DiT block. However, it struggles to maintain temporal consistency across video frames. By contrast, Scale-Adapter takes only a single condition while still being capable of both multi-condition and zero-shot learning.

## 3 PRELIMINARIES

**Latent Diffusion Models** Many recent video generation works utilize latent diffusion models (LDMs) (Rombach et al., 2022) to learn the compact representations of videos. First, given a $F$-frame RGB video $\boldsymbol{x} \in \mathbf{R}^{F \times 3 \times H \times W}$, a video encoder (of a pretrained autoencoder) provides $C$ dimensional latent representation (latents):

$$\boldsymbol{z} = \mathcal{E}(\boldsymbol{x}) \in \mathbf{R}^{F \times C \times H' \times W'}, \tag{1}$$

where height and width are spatially downsampled ($H' < H$ and $W' < W$). Next, in the forward process, a noise scheduler (e.g., DDPM (Ho et al., 2020)) adds noise to the latents $\boldsymbol{z}$. Then, in the backward pass, a diffusion model $\mathcal{F}_{\boldsymbol{\theta}}(\boldsymbol{z}_t, t, \boldsymbol{c}_{\text{text/img}})$ learns to gradually denoise the latents, given a diffusion timestep $t$, and a text prompt $\boldsymbol{c}_{\text{text}}$ (T2V) and/or an initial frame $\boldsymbol{c}_{\text{img}}$ (I2V) if provided. The diffusion model is trained with the objective: $\mathcal{L}_{\text{LDM}} = \mathbf{E}_{\boldsymbol{z}, \boldsymbol{\epsilon} \sim N(0, \boldsymbol{I}), t} \|\boldsymbol{\epsilon} - \boldsymbol{\epsilon}_{\boldsymbol{\theta}}(\boldsymbol{z}_t, t, \boldsymbol{c}_{\text{text/img}})\|_2^2$, where $\boldsymbol{\epsilon}$ and $\boldsymbol{\epsilon}_{\boldsymbol{\theta}}$ represent the added noise to latents and the predicted noise by $\mathcal{F}_{\boldsymbol{\theta}}$ respectively. We apply the same objective for the adapter methods training.

**ControlNets** ControlNet (Zhang et al., 2023) is designed to add spatial controls (e.g., depth, sketch, segmentation maps) to image diffusion models. Specifically, given a pretrained backbone image diffusion model $\mathcal{F}_{\boldsymbol{\theta}}$ that consists of input/middle/output blocks, ControlNet has a similar architecture $\mathcal{F}_{\boldsymbol{\theta}'}$, where the input/middle blocks parameters of $\boldsymbol{\theta}'$ are initialized from $\boldsymbol{\theta}$, and the output blocks consist of $1 \times 1$ convolution layers initialized with zeros. ControlNet takes the diffusion timestep $t$, text prompt $\boldsymbol{c}_{\text{text}}$, control image $\boldsymbol{c}_{\text{f}}$, and the noisy latents $\boldsymbol{z}_t$ as inputs, and the output features are merged into the backbone model $\mathcal{F}_{\boldsymbol{\theta}}$ for final image generation. Unlike the U-Net architecture, DiT-based ControlNet requires training and copying the entire DiT block, including extra spatial compression with time embedding and a zero-init module, which increases the number of parameters.

## 4 METHOD

### 4.1 TASK DEFINITION

Given a text description $T$, diverse visual conditions $C$, large text-to-video diffusion model $F_l$, conditional small video diffusion model $F_s$, the goal of Scale-Adapter $S$ is to transfer various control signal guided generation ability in $F_s$ to $F_l$ without additional ControlNet training. A core requirement of $S$ is that $V_{\text{gen}}$ aligns with both text description $T$ and diverse visual conditions $C$. Formally, this conditional video generation task is formulated as:

$$V_{\text{gen}} = F_l \left( T, S(F_s(C)) \right). \tag{2}$$

Our designs are detailed in subsequent sections: Section 4.2 presents the overall architecture of $\mathcal{F}$, outlining the interaction mechanism between the Small Video Diffusion Model, Scale-Adapter, and Large Video Diffusion Model. Section 4.3 describes the details of adapter design, which enables efficient transfer of control conditions across different model scales.

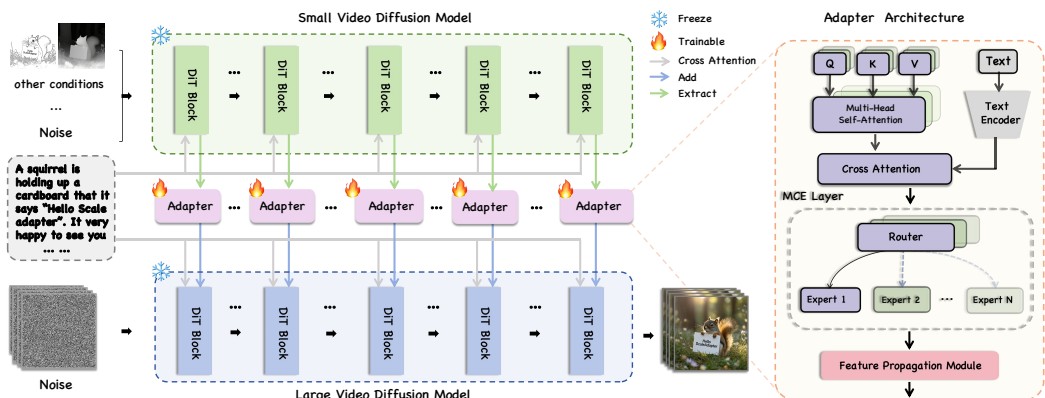

Figure 4: **Overview of Scale-Adapter**. **Left**: To drive a large text-to-video diffusion model with new conditions, we first feed the condition latents to a frozen small pretrained conditional diffusion model, whose features are first injected into Scale-Adapter and then mapped to the frozen large diffusion model. **Right**: For each adapter, we design an *Mixture of Condition Experts* (MCE) layer to learn multiple control signals and a Feature Propagation module to transfer knowledge efficiently.

## 4.2 SCALE-ADAPTER TRAINING STRATEGY

The framework in Figure 4 illustrates our methodology for enabling efficient transfer of scalable multi-condition control knowledge. Scale-Adapter propagates the knowledge from a small video diffusion model to a large one, keeping all parameters of both pretrained models frozen. Notably, the small video diffusion model, initialized from a pre-trained text-to-video diffusion model, requires prior fine-tuning or LoRA training (Hu et al., 2022) to adapt to multiple conditions. Since the large video diffusion model has a different number of DiT blocks than the small model, we select the first, last, and several middle DiT blocks to transfer conditional latent features to the large model. This design allows us to train only the Scale-Adapter, resulting in significantly higher efficiency compared to fine-tuning the large model itself.

## 4.3 ADAPTER ARCHITECTURE

**Cross-Scale Knowledge Bridging**   Inspired by ControlNet, conditional information is effectively injected into the target backbone using trainable copies of the diffusion model block and zero-initialized linear layers. As illustrated in Figure 4, our Scale-Adapter for the DiT-based video diffusion model employs a novel architecture consisting of three key components: attention modules, *Mixture of Condition Experts*(MCE), and a *Feature Propagation Module*. Leveraging the MCE layer to dynamically route condition tokens and adapt the timestep embedding $t$ derived from the small video diffusion model within the Feature Propagation Module, our design ensures consistent conditional and temporal representation throughout the bridging stage of the diffusion process. At the same time, consider increasing the minimum number of additional parameters.

**Mixture of Condition Experts**   Observing that intrinsic connections exist among different conditioning signals, such as between canny edges and depth maps shown in Figure 3, we are inspired to develop a unified architecture that leverages these relationships for multi-condition generation. To avoid the inefficiency of retraining for each new condition while ensuring high scalability and zero-shot adaptation capability to unseen conditions, we introduce a *Mixture of Condition Experts* (MCE) layer. This module comprises a specialized set of experts within Scale-Adapter that work together to capture and integrate latent features from diverse conditional inputs, such as depth maps, canny edges, and human poses. Within the MCE layer, different experts are designed to simultaneously learn from various conditional signals. Only a sparse subset of experts is activated during processing, enabling effective and efficient fusion under both single and multi-condition settings. This structure further allows the model to exhibit zero-shot generalization to new, unseen conditions during inference. Moreover, the MCE layer offers high extensibility. When introducing a new condition or task, new experts can be seamlessly added. These experts can be initialized using weights shared from existing experts, facilitating rapid convergence with minimal training data.

Our MCE layer consists of two types of parameterized experts: shared experts $\mathcal{E}_s$ and condition-specific experts $\mathcal{E}_c$. This design enables zero-shot generalization to unseen conditions by leveraging knowledge from related expert modules. Mathematically, given a set of $K$ feature tokens $\{c_1, c_2, \ldots, c_K\}$, the MCE layer computes the conditional output $h_t^{mce}$ in timestep $t$ as:

$$h_t^{mce} = \sum_{k=1}^{K} g_k(c_k, t) \cdot \mathcal{E}_{c_k}(x_t^a, t), \qquad (3)$$

where $g_k(c_k, t) = \text{Softmax}(\text{MLP}_g([c_k; t]))$ denotes the gating function that assigns weights to each expert $\mathcal{E}_{c_k}$ based on the input condition $c_k$ and $x_t^a$ is adapter attention module output. The shared expert $\mathcal{E}_s$ is integrated via:

$$\mathcal{E}_{c_k}(x_t^a, t) = \mathcal{E}_s(x_t^a, t) + \Delta\mathcal{E}_{c_k}(x_t^a, t), \qquad (4)$$

where $\Delta\mathcal{E}_{c_k}$ represents the condition-specific adaptation parameters. During inference, only the relevant experts are activated via dynamic routing, reducing computational overhead. Empirically, this design achieves state-of-the-art multi-condition synthesis with fewer parameters compared to naive Multi-ControlNet baselines while maintaining condition and temporal consistency across frames.

**Feature Propagation**    To efficiently transfer condition information from the large model to the small model, we introduce a *Feature Propagation Module* that includes a learnable modulation factor, a time projection layer, and an Up-Projection layer. The Up-Projection layer leverages a linear layer to transfer condition information from the small video diffusion model to the large video diffusion model. Then, the learnable scaling modulation with a time projection layer dynamically adapts condition features into the large video diffusion model. Those design enables Scale-Adapter to modulate latent feature contributions based on the denoising stage adaptively. Specifically, given the adapter's latent feature $x_t^a$ at timestep $t$, we compute the cross-attention output $y$ between $x_t^a$ and the text embedding $c_{txt}$. The feature propagation process is formalized as follows:

$$\begin{aligned} \alpha_{\text{scale}} &= \text{Modulation} + \text{Time\_Projection}(t), \\ x_t^a &= \text{Up\_Projection}(x_t^a) \cdot \alpha_{\text{scale}} + \text{Up\_Projection}(h_t^{mce}), \end{aligned} \qquad (5)$$

Where $\alpha_s$ denotes a learnable scale modulation factor that the latent feature transfers to the target video backbone. The overall feature propagation function is defined as

$$x_t^{a'} = u(x_t^a, c_{txt}, t; \theta), \qquad (6)$$

where $x_t^{a'}$ represents the adapter output and $\theta$ encompasses the adapter's trainable parameters. The scale features are then integrated into the large model's latent space via:

$$x_t = x_t + x_t^{a'}, \qquad (7)$$

Where $x_t$ is the latent large video diffusion model during the denoise stage. This additive integration ensures that the large model's prior knowledge is augmented with condition-specific information while preserving its structural integrity. Through this module, Scale-Adapter effectively bridges the gap between small, specialized models and large foundation models, enabling efficient knowledge transfer across scales. Empirically, it reduces trainable parameters by 70% (without MCE layer) compared to DiT-ControlNet while maintaining comparable performance.

## 5    EXPERIMENTS

### 5.1    IMPLEMENTATION DETAILS

Scale-Adapter integrates Diffusion Transformer Blocks with Multiple Condition Experts (MCE) Layer (Shazeer et al., 2017). We conduct experiments using two open-source text-to-video diffusion models as backbones: Wan2.1-1.3B and Wan2.1-14B (Wan et al., 2025), as well as CogVideoX-2B and CogVideoX-5B (Hong et al., 2022). Training required approximately 2 days on 1×NVIDIA H100 80GB GPU. We sampled 15K videos from the Koala-36M dataset (Wang et al., 2024a) and generated degraded versions by converting samples to grayscale and downscaling to low resolution. Before training, we extracted auxiliary conditioning signals (human pose, depth maps, and Canny edges) from all videos. For evaluation, we manually curated 100 high-quality videos spanning diverse content categories. For conditional generation tasks involving reference videos, we report LPIPS (Zhang et al., 2018), SSIM (Wang et al., 2004), CLIP Score(semantic correspondence between generated and reference content), and FVD metrics (Unterthiner et al., 2018).

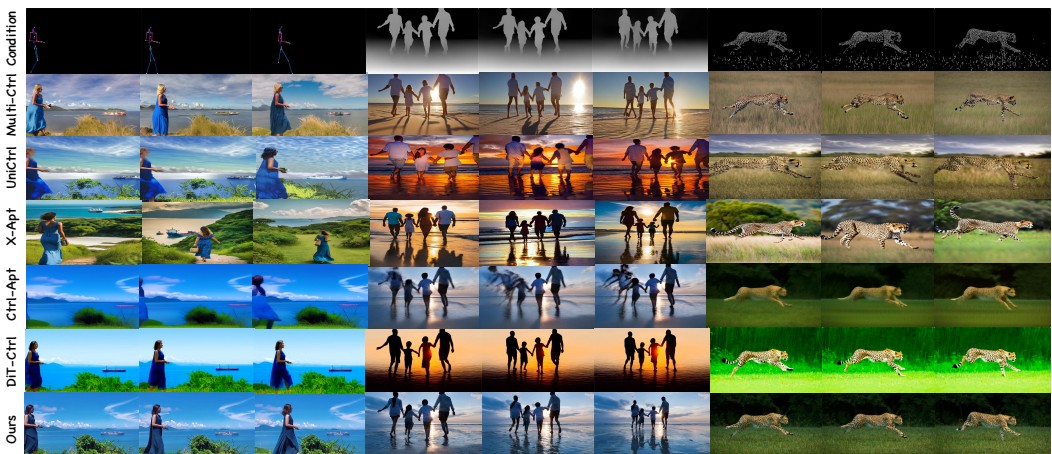

Figure 5: **Qualitative comparisons with baselines.** "Ctrl" is "ControlNet" and "Apt" is "Adapter." We perform the visual comparison with five baselines using the same conditions, while the image-based method shows poor performance in cross-frame consistency, and our method obtains better performance in the adapter-based methods.

## 5.2 QUALITATIVE RESULTS

We visually compare the performance of our method against baseline models across three key conditions (Canny edges, Depth maps, and Openpose skeletons) in Fig. 5. Our approach consistently outperforms alternatives in both visual quality and alignment with input conditions and text prompts, as validated by the qualitative examples. Under pose control, our method achieves significantly tighter spatial alignment with input skeletons compared to baselines. For the prompt "a woman walking along the shoreline" on Fig. 5, our results accurately adhere to pose constraints while maintaining natural motion. In contrast, UniControl and Uni-ControlNet misinterpret skeletal configurations. For instance, the blue-dressed woman in their outputs exhibits inconsistencies in appearance and motion coherence, with subtle frame-to-frame discrepancies undermining temporal consistency. Our model, by contrast, preserves precise pose adherence while ensuring smooth, natural movements.

For depth control generation, our framework demonstrates a superior understanding of 3D geometry, producing outputs with geometrically plausible structures from depth maps and surface normals. Ctrl-Adapter, by comparison, exhibits noticeable geometric inconsistencies, such as distorted character proportions and implausible spatial relationships between objects. The X-Adapter captures basic human semantics but suffers from significant inter-frame variations in character appearance, lacking the temporal consistency necessary for video generation. Our method, however, maintains both geometric fidelity and cross-frame coherence. For edge-guided generation, our model outperforms ControlNet-based methods in edge preservation and structural consistency. As shown in Fig. 5,

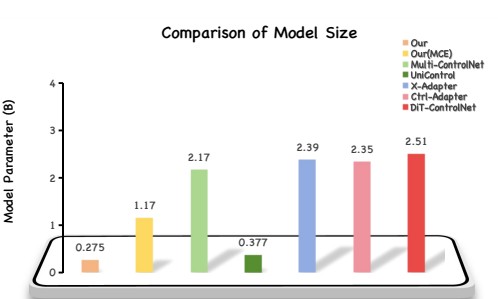

Figure 6: **Comparison of trainable model parameters in the diffusion model**. Our methodology requires the fewest trainable parameters.

this advantage is particularly pronounced in motion details, for example, the leg movements of the yellow cheetah in the examples, where competing methods exhibit noticeable blurring or edge misalignment. Our results retain sharp, faithful alignment with input edges while preserving the fluidity of dynamic motions. These qualitative findings reinforce that the integrated design of our method, combining the condition fusion of the MCE layer and efficient feature propagation, better balances condition adherence, visual quality, and temporal consistency between various control signals.

Table 1: Comparison of State-of-the-Art Baselines. The best result in each column is **bolded**, and the second best is underscored.

| Model | Canny Edge | | | | Depth Map | | | | Pose | | | | Temproal Consistency ↑ |
|---|---|---|---|---|---|---|---|---|---|---|---|---|---|
| | FVD↓ | CLIP↑ | LPIPS↓ | SSIM↑ | FVD↓ | CLIP↑ | LPIPS↓ | SSIM↑ | FVD↓ | CLIP↑ | LPIPS↓ | SSIM↑ | |
| X-Adapter (Ran et al., 2024) | - | 0.545 | 0.736 | 0.209 | - | 0.517 | 0.759 | 0.127 | - | - | - | - | 0.754 |
| Uni-ControlNet (Zhao et al., 2024) | - | 0.642 | 0.575 | 0.322 | - | 0.531 | 0.778 | 0.214 | - | 0.509 | 0.823 | 0.188 | 0.763 |
| UniControl (Qin et al., 2023) | - | 0.584 | 0.773 | 0.268 | - | 0.572 | 0.791 | 0.178 | - | 0.541 | 0.741 | 0.207 | 0.876 |
| Ctrl-Adapter (Lin et al., 2024) | 2135.302 | 0.757 | 0.358 | 0.619 | 2241.457 | 0.785 | 0.352 | 0.616 | 2437.149 | 0.712 | 0.672 | 0.304 | 0.981 |
| DiT-ControlNet (Denis, 2025) | 2126.246 | 0.781 | 0.551 | 0.369 | 2702.865 | 0.729 | 0.686 | 0.304 | 2685.619 | 0.645 | 0.777 | 0.295 | 0.978 |
| Wan2.1-14B(fine-tuned) | **1145.931** | 0.919 | **0.187** | 0.675 | **1271.194** | 0.912 | **0.193** | **0.664** | **1004.556** | **0.926** | **0.161** | **0.691** | 0.979 |
| Ours | 1447.829 | **0.918** | 0.255 | **0.585** | 1461.706 | **0.913** | 0.251 | 0.591 | 1502.910 | 0.903 | 0.273 | 0.573 | **0.984** |

## 5.3 QUANTITATIVE RESULTS

We conducted comprehensive comparisons against state-of-the-art ControlNet-based and Adapter-based methods. As shown in Table 1, Scale-Adapter, deployed on the 14B-T2V base model, out-performs existing strong video control methods across both depth map and Canny edge conditions, achieving competitive performance in visual quality and similarity to the reference video. For Adapter-based methods, our model outperforms X-Adapter and Ctrl-Adapter across most metrics, despite notable differences in training resources: while these baselines utilize datasets of over 100K videos or images and fewer GPUs, our method is trained on a more compact 10K video dataset. Notably, our base model lacks prior conditioning capabilities, highlighting the efficiency of our adapter design in injecting control capabilities into pre-trained text-to-video models. For ControlNet-based methods, we further compared against UniControl and Uni-ControlNet. We keep their method setting using the image diffusion model to generate frames. For DiT-ControlNet, we trained separate ControlNets for each condition as a baseline. Scale-Adapter achieves the competitive FVD, LPIPS, and CLIP scores. In contrast, existing ControlNet-based approaches require distinct ControlNets for different conditions, allowing specialized training for each task, yet our method still achieves overall competitive performance. Additionally, we provided a comparative experiment of the same model architecture for small models, large models, and our method in different tasks in the supplementary materials. These results validate that Scale-Adapter's architecture delivers great reversed distillation ability for conditional video generation, balancing efficiency and performance across diverse metrics.

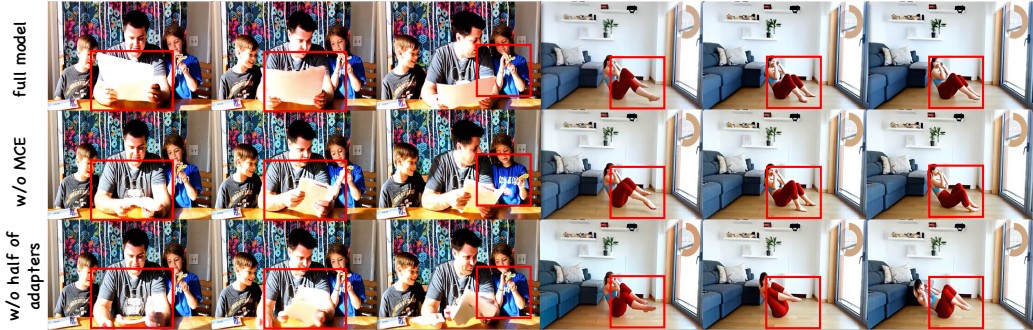

Figure 7: **Ablation results**. We present results by removing the MCE layer and changing the number of adapters. Without the MCE layer and a half number of adapters, it exhibits different levels of degradation in motion coherence and quality.

## 5.4 ABLATION STUDY

We systematically evaluate the core components of Scale-Adapter through controlled ablation experiments, utilizing five metrics: FVD, LPIPS, SSIM, and CLIP Score. Quantitative comparisons in Table 2 reveal two key insights: **MCE Layer Efficacy.** The full model with the *Mixture of Condition Experts* (MCE) layer achieves an FVD improvement across metrics compared to MCE-ablated variants, underscoring the critical role of dynamic condition feature fusion. This improvement stems from mixed training on multiple conditions, which enables the model to learn intrinsic relationships between different conditions. While the MCE-ablated variant retains basic conditioning control, it exhibits measurable degradation in motion coherence. As shown in Figure 7, the MCE-equipped model produces clearer and smoother details in video characters (*e.g.*, hands and legs). Moreover, in

scenarios with multiple characters (left panel of Figure 7), the MCE layer better controls individual character motions and their interactions with objects.

**Adapter Scaling Efficiency.** As depicted in Figure 6, reducing the number of adapters from 12 to 7 maintains robust conditioning fidelity; most metrics show no significant degradation, while reducing adapter parameters by nearly half. This indicates that the combined design of the Feature Propagation Module and MCE layer effectively mitigates performance drops even with fewer

| Configuration | FVD ↓ | LPIPS ↓ | SSIM ↑ | CLIP ↑ |
|---|---|---|---|---|
| Full Model | **1461.706** | **0.251** | **0.591** | **0.913** |
| w/o MCE | 1516.011 | 0.268 | 0.573 | 0.904 |
| w/o Half of adapters | 1990.068 | 0.355 | 0.567 | 0.875 |

Table 2: Ablation Study of Key Components

adapters, making the model less sensitive to adapter count. However, further reducing the number of adapters leads to noticeable declines in video quality, particularly in condition consistency. These results demonstrate that the synergistic integration of the MCE layer and adaptive adapter scaling achieves state-of-the-art efficiency performance tradeoffs. This design enables precise conditional control with only single-pass inference, balancing model lightness and control capability. The additional zero-shot quantitative experiment is on Tab. 4 and Tab. 3 in the supplementary materials.

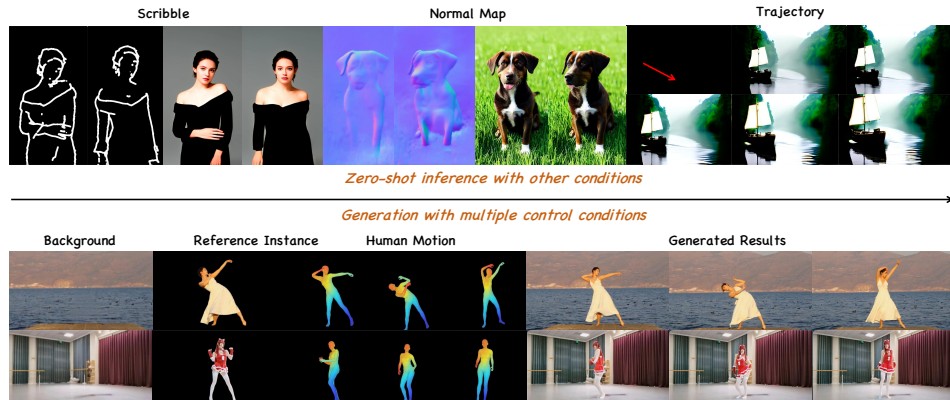

Figure 8: **Gallery of our proposed methods**. Given multiple conditions, including background, reference instance, and human motion, our method has the capability of transferring control conditions and unseen conditions to generate a high-quality and controllable video clip.

## 6  APPLICATIONS AND DISCUSSION

**Zero-Shot generalization**. Scale-Adapter serves as a knowledge bridge that maps diverse control conditions into the unified representation space of the backbone generative model. To evaluate its generalization capability, we directly apply Scale-Adapter to large video diffusion models using condition types not seen during training. The Quantitative experiments are provided in Sec. D in the appendix. As illustrated in Fig. 8, the model successfully handles scribble, normal map, and trajectory conditions despite their absence in the training set, demonstrating that the adapter facilitates zero-shot conditional extension through knowledge transfer between foundation models. **Unified multiple condition synthesis**. Our method supports the simultaneous integration of heterogeneous conditional signals, as shown in Fig. 1 and Fig. 8. Scale-Adapter dynamically routes these multimodal inputs through condition-specific pathways while maintaining coherence across all control modalities.

## 7  CONCLUSION

This paper has proposed *Scale-Adapter*, a novel reversed distillation adapter for efficiently transferring conditional information from small adapted models to large video diffusion transformers. Our method achieves robust performance in single and multi-condition video generation with significantly reduced training costs. Despite these advantages, our approach occasionally struggles to render fine-grained details. Future work will focus on enhancing spatial-temporal representation learning to improve fidelity in dynamic regions.

## ETHICS STATEMENT AND REPRODUCIBILITY

All procedures performed in studies involving human participants were in accordance with the ethical standards of the institutional and/or national research committee. This article does not contain any studies with human participants performed by any of the authors. Informed consent was obtained from all individual participants included in the study. We will release code under **CC-BY-NC-4.0**.

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

APPENDIX

## A LATENT DISTRIBUTION STATISTIC ANALYSIS

To evaluate the transferability of feature distributions between the large and small models, we randomly selected a single data sample as input and conducted a statistical analysis of the latent spaces in the Diffusion Transformer Blocks of both models. We pair the DiT Blocks of the 1.3B model and the 14B model according to a specific sequence (Wan2.1 14B-T2V block index: [0, 4, 8, 12, 16, 20, 24, 28, 32, 36]; Wan2.1-1.3B(fine-tuned from 1.3B-T2V) block index: [0, 2, 5, 8, 11, 14, 15, 18, 21, 24]), and then conduct statistical analysis.

The analysis metrics were divided into two components: (1) Post-PCA dimensionality reduction: Statistical measures including mean correlation, standard deviation correlation, covariance similarity, Wasserstein distance, etc.; (2) Calculation of direct distribution distance: Metrics calculated without dimensionality reduction, such as Wasserstein distance and matrix similarity. Results from this

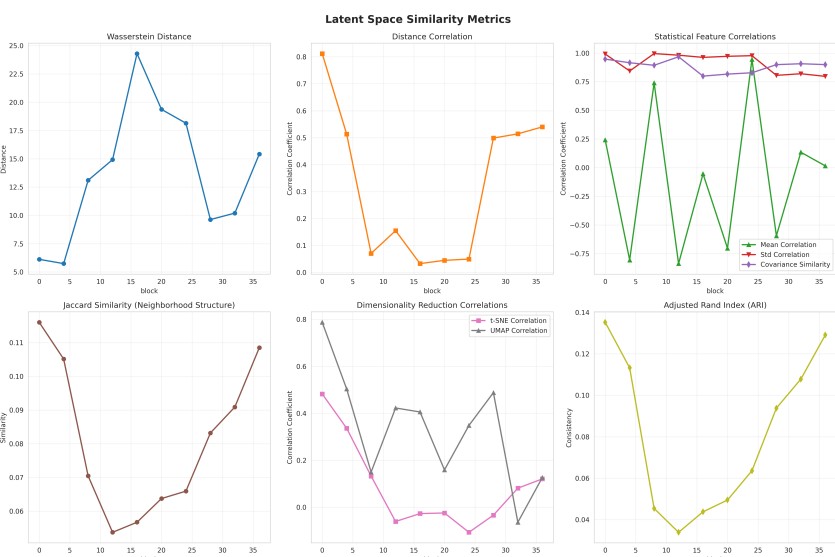

Figure 9: **Latent feature similarity between two scale models**. Observing the latent features of the first few and last few DiT Blocks in the two models exhibit high similarity, whereas the latent features of the middle Blocks show relatively low similarity. The two models exhibit volatility differences at different network levels, but the ultimately generated latent space representations have global consistency.

analysis are presented in Figure 9. We first analyzed the metrics from the paired DiT Blocks to characterize the relationship between the two latent space distributions. While distributional similarity varied substantially across blocks (e.g., Wasserstein distance ranged from 5.7 to 24.3, and distance correlation values spanned 0.03 to 0.81), classifier accuracy remained consistently close to 0.5 (range: 0.458–0.472) across all blocks. This observation indicates that although the local statistical properties of the two latent spaces differ, their overall representations are highly analogous and indistinguishable by a classifier across all model blocks. From our analysis results, we observe that the latent features of the first few and last few DiT Blocks in the two models exhibit high similarity, whereas the latent features of the middle Blocks show relatively low similarity. Inspired by this phenomenon, the Adapters in Scale-Adapter are primarily concentrated in the first few and last few Blocks, while the middle Blocks are equipped with fewer Adapters that are evenly spaced. This design is intended to effectively facilitate the transfer of features from the small model to the large model.

In conclusion, our analysis of the paired DiT Blocks reveals that the latent space distributions of the two models are generally analogous, though their similarity varies substantially across blocks: some blocks exhibit strong similarity, while others show notable divergence. For the feature transfer task, a block-by-block strategy is therefore well-suited—blocks with high similarity metrics are more amenable to direct transfer or distillation. Building on this insight, our work aims to develop a method

for adapting features across DiT Blocks between small and large video diffusion models to enhance the efficacy of reversed distillation.

## B  ADDITIONAL ARCHITECTURE DETAIL

As shown in Figure 9, the core architecture of Scale-Adapter consists of three main components: an attention module, a Mixture of Condition Experts (MCE) layer, and a Feature Propagation Module.

The attention module is a fundamental component within each transformer block, comprising three key sub-modules: LayerNorm layers, a self-attention mechanism, and a cross-attention mechanism. The LayerNorm layer first normalizes the input features to stabilize training. The self-attention mechanism then captures contextual dependencies among spatial and temporal tokens. Finally, the cross-attention module integrates conditional information (such as text or structural guidance) into the visual representation. This design enables effective fusion of spatial, temporal, and conditional features throughout the diffusion process.

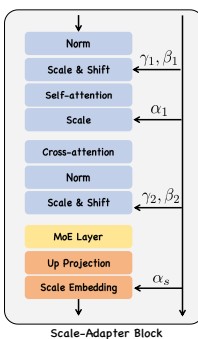

Figure 9: **Adapter detail**. Two important principles of design are multi-condition integration and low-resource efficient training.

The MCE layer includes the router, experts, and shared experts. Each expert is made up of an MLP layer. The default setting has 1 shared expert and 3 task-specific experts, and the topk weight is 2.

The Feature Propagation Module consists of a scale embedding layer and an up-projection layer. The scale embedding layer includes a scale factor $\alpha_s$ and a time embedding layer that can efficiently achieve the transmission of control condition information. The time embeddings of other layers are all initialized from the small model. These low-dimensional features encapsulate condition-specific information and enable seamless integration with the pre-trained small model. To support efficient training and inference, the MCE layer can be optionally removed, reducing the total parameter count by up to 50% without compromising performance. The Feature Propagation Module comprises an up-projection layer and a scale embedding mechanism, which together facilitate efficient and robust transfer of conditional features to the large diffusion model.

## C  TRAINING SETTING AND DATA PROCESS

Our model is trained with the following key hyperparameters: a learning rate of $2 \times 10^{-5}$, a training batch size of 2, a video sampling strategy that selects 24 frames per video with a sampling stride of 2, a video sample resolution of 512×512 (denoted as video sample size=512), a token sequence length of 512 (denoted as token sample size=512), 1 gradient accumulation step to stabilize training, and mixed-precision training using the "bf16" (bfloat16) format to balance computational efficiency and numerical precision Chen et al. (2023); Chowdhery et al. (2022); Yang et al. (2024); Chen et al. (2025); Ramesh et al. (2022); Li et al. (2023a).

For condition data preparation, we leverage three conditions extracted from input frames: depth maps obtained via Depth-Anything (a state-of-the-art monocular depth estimation model), structural boundaries extracted using a Canny edge detector, and human pose skeletons retrieved through the OpenPose framework, ensuring the model captures both global scene geometry and fine-grained semantic details (Ryu, 2022; Dai et al., 2023a; Li et al., 2017; Ma et al., 2024; Gal et al., 2023; Goodfellow et al., 2020; Parmar et al., 2022).

## D  ZERO-SHOT WITH UNSEEN CONDITIONS

Our method demonstrated strong zero-shot generalization ability after a small number of conditional adaptation training. Our model is trained exclusively on conditional video data, including depth,

Table 3: Zero-shot Generation Performance on Unseen Conditional Inputs

| Conditional | FVD (↓) | LPIPS (↓) | SSIM (↑) | CLIP (↑) |
|---|---|---|---|---|
| Canny(trained) | 1447.82 | 0.25 | 0.59 | 0.92 |
| Normal Map | 1919.29 | 0.26 | 0.51 | 0.86 |
| Scribble | 1752.51 | 0.25 | 0.52 | 0.87 |
| Segmentation Map | 1925.24 | 0.27 | 0.49 | 0.86 |
| MLSD | 1966.51 | 0.25 | 0.51 | 0.86 |
| Line Art | 1655.78 | 0.23 | 0.52 | 0.88 |

Table 4: Experiment Between Different Scale Models

| Model | FVD (↓) | CLIP (↑) | LPIPS (↓) | SSIM (↑) |
|---|---|---|---|---|
| Wan-Control-1.3B(T2V) | 2819.091 | 0.785 | 0.615 | 0.337 |
| Wan-Control-14B(T2V) | **2505.551** | **0.801** | **0.576** | **0.351** |
| Ours(T2V) | 2797.460 | 0.786 | 0.629 | 0.341 |
| Wan-Control-1.3B(I2V) | 1563.066 | 0.896 | 0.205 | 0.572 |
| Wan-Control-14B(I2V) | **1271.194** | 0.913 | **0.193** | **0.664** |
| Ours(I2V) | 1516.011 | **0.914** | 0.218 | 0.584 |

pose, and Canny conditions. To evaluate its zero-shot generalization capability, we test the model on several previously unseen conditional inputs, including normal maps, scribbles, segmentation maps, MLSD edges, and line art.

Quantitative results are summarized in Table 3, where our method is compared against existing approaches using widely adopted metrics including FVD, LPIPS, SSIM, and CLIP score. The results demonstrate that our approach effectively adapts to novel conditions without additional fine-tuning, maintaining high visual quality and semantic alignment across diverse control signals. The experiment results demonstrate that our MCE layer not only supports multi-condition adaptation but also zero-shot learning.

## E  ADDITIONAL EXPERIMENT

Our method is also competitive compared to models that have undergone large-scale pre-training under the same architecture. It is noted that both our large and small models are initialized from text-to-video models, with training data of less than 100k and a time of less than 48 GPU hours. We have conducted additional experiments comparing the performance of our method against 1.3B and 14B parameter models on both Image-to-Video (I2V) and Text-to-Video (T2V) generation tasks. For the baseline models, we used the pre-trained Wan2.1-Fun-Control model with 1.3B and 14B parameters, respectively. Our approach is based on the Wan2.1-14B-T2V model, with depth maps consistently applied as the control condition. The I2V test set comprises 100 samples selected from the Koala-36M dataset (Wang et al., 2024a), whereas the T2V evaluation utilizes 1,000 samples from the Panda-70M dataset (Chen et al., 2024b).

As shown in Table 4, experimental results demonstrate that our method achieves comparable performance to the heavily trained 1.3B and 14B models in the I2V task, and even outperforms the 14B model in the T2V task, demonstrating highly competitive generation quality. These results confirm that our approach effectively transfers knowledge from the small conditional model to the large base model, achieving strong performance with greater parameter efficiency.

## F  THE USAGE OF LARGE LANGUAGE MODELS

In this paper, the usage of the LLM mainly falls into the following aspects: specifically, for grammar checking and format optimization, we use DeepSeek-R1 to conduct grammar error checking on the paragraphs of the paper as well as format checking of charts and graphs; additionally, for language polishing, we apply Doubao to polish and optimize the language expression of the paper's text

description part; and it is important to note that all authors are responsible for the content generated by the LLMs.

