# OpenReview forum: "Scale-Adapter: Reversed Distillation Adapter for Efficient Training of Large Video Diffusion Models"
_ICLR.cc/2026/Conference — ICLR 2026 Conference Withdrawn Submission_

### Official Review · Reviewer_omSh · 2025-10-27

**Soundness:** 2
**Presentation:** 2
**Contribution:** 2
**Rating:** 2
**Confidence:** 4

**Summary:**

The authors introduce Scale-Adapter, a plug-and-play module for efficient and flexible training of large video diffusion models under diverse conditional controls. The method leverages a reversed distillation strategy that transfers conditional knowledge from small, pre-trained video diffusion models that can handle conditions to large frozen video models, by learning adapter layers that bridge them. A key component is the Mixture of Condition Experts (MCE) layer, which enables multi-condition control via dynamic routing within a unified architecture. Another innovation is the Feature Propagation Module, ensuring coherent and temporally aligned feature transfer. Experiments are conducted using Wan2.1 and CogVideoX backbones on the Koala-36M dataset with Canny, pose, and depth conditions. The method shows competitive or superior performance in both qualitative and quantitative evaluations (FVD, LPIPS, SSIM, CLIP), particularly excelling in temporal coherence and multi-condition fusion.

**Strengths:**

1. **Computational Efficiency.** The proposed method performs knowledge transfer, using a small model to inject conditional control into a large frozen model via adapters. This avoids the computationally expensive full-model fine-tuning.
2. **Strong Empirical Results.** The model achieves competitive results on FVD, CLIP, and LPIPS across multiple tested conditions, while requiring fewer trainable parameters. It also shows robust performance in ablation and zero-shot generalization experiments.

**Weaknesses:**

1. **No Same-Backbone Adapter Baseline Comparison.** The authors report results of Scale-Adapter on Wan2.1 and CogVideo-X. However, it’s not clear if they re-implement baseline methods such as Ctrl-Adapter on the same backbone architecture for fair comparison. This weakens claims of efficiency and performance gains relative to prior work that carefully controlled for this factor.
2. **Method novelty.** The use of pre-trained conditional feature generator to guide downstream image/video diffusion models has been already studied in X-Adapter and Ctrl-Adapter. Mixture of Condition Experts (MCE) module bears strong resemblance to the MoE-style adapter routing mechanism proposed in Ctrl-Adapter, but this similarity is not discussed.
3. **Unclear Explanation of Parameter Efficiency in Figure 6.** It is not clearly explained how the proposed method achieves fewer trainable parameters compared to previous methods like X-Adapter and Ctrl-Adapter. The mechanism or architectural difference leading to this reduction should be better justified in the text.
4. **Lack of Detail on Small Teacher Model.** The paper references a set of fine-tuned small video diffusion models as “teachers” for reversed distillation (e.g., Figure 3), but does not specify what models were used, how many per condition, or how they were trained. It also remains unclear whether such small video models exist at scale across many condition types.

**Questions:**

- Citation mismatch in efficiency reporting: The paper refers to Figure 6 when discussing parameter efficiency of the MCE layer, but the actual comparison appears in Table 2.

**Details Of Ethics Concerns:**

While checking the related work, I found the Sec. 3 (“Preliminaries”) contains text that closely mirrors Sec. 3.1 of Ctrl‑Adapter (https://arxiv.org/abs/2404.09967) with near‑identical paragraphs, raising concerns of uncredited text reuse and unclear attribution of prior work.

---

### Official Review · Reviewer_4G55 · 2025-10-31

**Soundness:** 2
**Presentation:** 3
**Contribution:** 2
**Rating:** 4
**Confidence:** 3

**Summary:**

This paper introduces Scale-Adapter, a plug-and-play adapter framework that efficiently transfers controllable knowledge from small video diffusion models to large-scale video diffusion transformers without requiring full fine-tuning.

**Strengths:**

1) Method. The paper presents a reversed knowledge distillation approach in diffusion models by transferring from smaller conditional models to larger pretrained video diffusion transformers. The method is highly efficient since it trains a single adapter to rule different conditions.
2) Experiements. The authors compare the method with other methods which also use a small video model as prior and achieves better performance.
3) Clarity. The paper is well-structured and easy to follow. Figures effectively visualize the problem motivation, method design, and results.

**Weaknesses:**

1) Method. The method is mainly adapted from x-adapter and ctrl-adapter, with an additional MCE module. It lacks insights and improvements on this kind of method. It would be better if the author can do a deep analysis on the feature propagation like which layer plays a more important rule and what is the feature alignment before and after the adapter is applied.
2) Metrics. The metrics the authors choose is not that proper. All the methods achieves >1000 FVD seems weird. There's also no metric to evaluate the condition fidelity.
3) Methods. The baselines the author choose is not enough and I will detail it in the questions.

**Questions:**

1) Method. Can the adapter be applied to other plugins like id-customization or stylization plugins other than controlnet?
2) Metrics. Is it proper to use FVD in the experiments? All the methods' FVD value are >1000 which is a quite high value. Also, can you provide a metric to evaluate the condition fidelity?
3) Experiments. Can you compare with other controllable video generation method like SparseCtrl[1] and VideoTetris[2] ?

[1] Guo, Yuwei, et al. "Sparsectrl: Adding sparse controls to text-to-video diffusion models." European Conference on Computer Vision.
[2] Tian, Ye, et al. "Videotetris: Towards compositional text-to-video generation." Advances in Neural Information Processing Systems 37 (2024): 29489-29513.

---

### Official Review · Reviewer_vnrD · 2025-10-31

**Soundness:** 3
**Presentation:** 1
**Contribution:** 3
**Rating:** 4
**Confidence:** 4

**Summary:**

This paper proposes an adapter-based architecture for structural control in video generation models.

**Strengths:**

The model is sufficiently lightweight, requiring minimal computational resources for both training and inference, while achieving satisfactory generation quality.

**Weaknesses:**

Controllable generation in video models can also be achieved via ControlNet, for example, as demonstrated in https://huggingface.co/alibaba-pai/Wan2.1-Fun-14B-Control . The authors should consider including such models as baselines in their experiments.

Some references are incorrectly cited—for instance, a “?” appears in line 147.

Additionally, some experimental results are mislabeled. For example, in line 385, the CLIP score of “Wan2.1-14B (fine-tuned)” is higher than that of “Ours”, yet the authors bolded “Ours”.

**Questions:**

Please refer to the weaknesses.

---

### Official Review · Reviewer_SBaG · 2025-11-02

**Soundness:** 3
**Presentation:** 2
**Contribution:** 3
**Rating:** 4
**Confidence:** 4

**Summary:**

The paper proposes Scale-Adapter, a method that uses a smaller diffusion backbone to simultaneously learn to encode multiple control signals and inject the control information to a larger diffusion model. The architecture relies on an adapter to transfer the information from the smaller control backbone to the larger generative model, which in turn comprises a MoE component and a additive component with learnable scale. Empirical results demonstrate better efficiency than controlnet approach and strong generalization abilities to new control modalities.

**Strengths:**

- The method demonstrates strong empirical results, significantly reducing trainable parameters while maintaining comparable results than more expensive methods (ControlNet, UniControl, Ctrl-Adapter, etc.).
- It shows impressive ability to handle unseen conditions without retraining, suggesting strong adaptability and scalability of the proposed architecture.

**Weaknesses:**

- The core method of the paper relies on the existence of a smaller pretrained video diffusion models that have the same number of layers (but smaller dimensions) as a larger video diffusion model. There may not be always a smaller version for state-of-the-art open-source models in the future, which limits the applicability of the proposed method. It would be great if the method can be applicable to the scenario where the small and large models are of different model family (e.g., CogVideo2B + Wan14B) and/or models of different number of layers.
- Minor issues:
  - Missing citation L147
  - In Figure 4 right, consider change the color scheme so the color of different experts are very different from green/blue - at the beginning I thought expert 1 corresponds to large video model and expert 2 corresponds to smaller conditional model due to the color similarity.
  - L297 alpha_s is not mentioned earlier, do you mean alpha_scale?
  - L296 shouldn't the output be x_t^{a^\prime} instead of x_t^a?

**Questions:**

The MCE design is motivated by the intrinsic connections exist among different conditioning signals. Are there any experiments that demonstrate that different experts actually become specialized to one (or a class of) modalities?

---

### Note · Authors · 2025-11-13

I have read and agree with the venue's withdrawal policy on behalf of myself and my co-authors.